# Sharing pain: Using pain domain transfer for video recognition of low grade orthopedic pain in horses

**Sofia Broomé**[1]*, **Katrina Ask**[2], **Maheen Rashid-Engström**[4,5], **Pia Haubro Andersen**[3], **Hedvig Kjellström**[1,6]

1 Division of Robotics, Perception and Learning, KTH Royal Institute of Technology, Stockholm, Sweden, 2 Department of Anatomy, Physiology and Biochemistry, Swedish University of Agricultural Sciences, Uppsala, Sweden, 3 Department of Clinical Sciences, Swedish University of Agricultural Sciences, Uppsala, Sweden, 4 Department of Computer Science, University of California, Davis, California, United States of America, 5 Univrses, Stockholm, Sweden, 6 Silo AI, Stockholm, Sweden

* sbroome@kth.se

**Data Availability Statement:** Data cannot be shared publicly because of confidentiality. DVM, Associate Professor Marie Rhodin (marie.

## Abstract

Orthopedic disorders are common among horses, often leading to euthanasia, which often could have been avoided with earlier detection. These conditions often create varying degrees of subtle long-term pain. It is challenging to train a visual pain recognition method with video data depicting such pain, since the resulting pain behavior also is subtle, sparsely appearing, and varying, making it challenging for even an expert human labeller to provide accurate ground-truth for the data. We show that a model trained solely on a dataset of horses with acute experimental pain (where labeling is less ambiguous) can aid recognition of the more subtle displays of orthopedic pain. Moreover, we present a human expert baseline for the problem, as well as an extensive empirical study of various domain transfer methods and of what is detected by the pain recognition method trained on clean experimental pain in the orthopedic dataset. Finally, this is accompanied with a discussion around the challenges posed by real-world animal behavior datasets and how best practices can be established for similar fine-grained action recognition tasks. Our code is available at https://github.com/sofiabroome/painface-recognition.

## 1 Introduction

Equids are prey animals by nature, showing as few signs of pain as possible to avoid predators [1]. In domesticated horses, the instinct to hide pain is still present and the presence of humans may disrupt ongoing pain behavior [2]. Further, recognizing pain is inherently subjective and time consuming, and is therefore currently challenging for both horse owners and equine veterinarian experts. An accurate automatic pain detection method therefore has large potential to increase animal welfare.

Orthopedic disorders are frequent in horses and are, although treatable if detected early, one of the most common causes for euthanasia [3–5]. The pain displayed by the horse may be subtle and infrequent, which may leave the injury undetected.

rhodin@slu.se) at the Swedish University of Agricultural Sciences can be contacted regarding the dataset, which can be made available for researchers who meet the criteria for access to confidential data. Data can also be requested from co-author, Katrina Ask (katrina.ask@slu.se).

**Funding:** Sofia Broomé is funded by the Swedish Research Council (https://www.vr.se), grant number 2016-03967 (award received by H.K. and P.H.A). Katrina Ask is funded by the Swedish Research Council FORMAS (http://www.formas. se), grant number 2016-01760 (MR) (award received by P.H.A). The funders had no role in study design, data collection and analysis, decision to publish, or preparation of the manuscript.

**Competing interests:** The authors have declared that no competing interests exist.

Pain is a complex multidimensional experience with sensory and affective components. The affective component is associated with changes of behaviour, to avoid pain or to protect the painful area [6]. While some of these behaviours may be directly related to the location of the painful area, such as lameness in orthopedic disorders and rolling in abdominal disorders [7], other pain behaviours, such as facial expressions, are thought to be universal as means of communication of pain with conspecifics. Acute pain has a sudden onset and a distinct cause, such as inflammation, trauma, or ischemia [8, 9] and all these elements may be present in orthopedic pain in horses.

Recognizing horse pain automatically from video requires a method for fine-grained action recognition, which can pick up subtle behavioral signals over long time, and further, the method should be possible to train using a small dataset. In many widely used datasets for action recognition [10–12], specific objects and scenery may add class information. This is not the case in our scenario, since the only valid evidence present in the video are poses, movements and facial expressions of the horse.

The Something-Something dataset [13] was indeed collected for fine-grained recognition of action templates, but its action classes are short and atomic. Although the classes in the Diving48 and FineGym datasets [14, 15] are complex and require temporal modeling, the movements that constitute their classes are densely appearing in a continuous sequence during the video, contrary to video data showing horses under orthopedic pain with sparse expressions thereof.

A further important complication is that the labels in the present scenario are inherently noisy, since the horse's subjective experience of pain can not be observed. Instead, pain induction and black /or human pain ratings are used as proxy when labeling video recordings. To further complicate matters, the behavioral patterns that we are searching for might appear in both pain and non-pain data, although with different frequency.

Expert pain assessment in horses is mainly performed by evaluating predetermined body behaviors and facial expressions displayed by the horse during a short observation period, usually two minutes. The observer can either stand outside the box stall or observe the horse in a video. Equine pain research has focused on identifying certain combinations of behaviors and facial expressions for general pain and specific types of pain, such as orthopedic pain [16–20]. It is important to understand that all behaviors and facial expressions are part of the non-verbal communication system of healthy horses as well, and that it is their combinations and frequency which can indicate if pain is present. Being an easier-to-observe special case, recordings of acute pain (applied for short duration and completely reversibly, under ethically controlled conditions) have been used to investigate pain-related facial expressions [21] and for automatic equine pain recognition from video [22]. Until now, it has not been studied how this generalizes to the more clinically relevant orthopedic pain.

This article investigates machine learning recognition of equine orthopedic pain characterized by sparse visual expressions. To tackle the problem, we use domain transfer from the recognition of clean, experimental acute pain, to detect the sparsely appearing visible bursts of pain behavior within low grade orthopedic pain data (Fig 1). We compare the performance of our approach to a human baseline, which we outperform on this specific task (Fig 2).

Our contributions are as follows:

- We are the first to investigate domain transfer between the recognition of different types of pain in animals.

- We present extensive empirical results using two real-world datasets, and highlight challenges arising when moving outside of clean, controlled benchmarking datasets when it comes to deep learning for video.

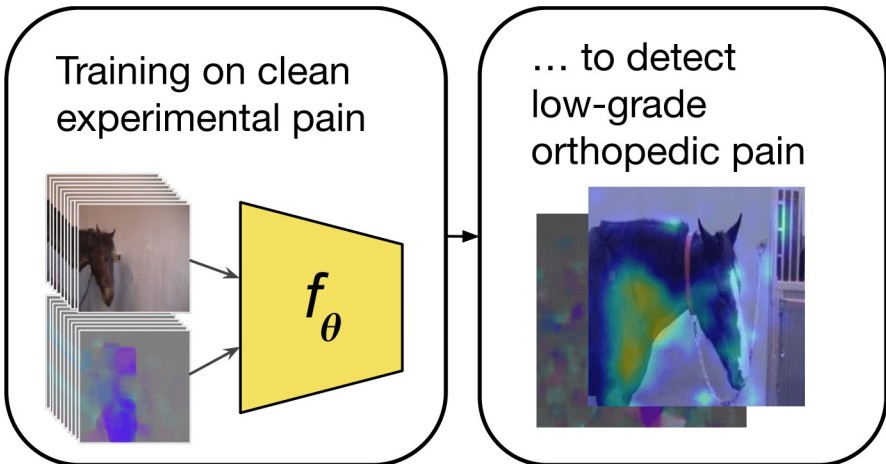

**Fig 1. We present a study of domain transfer in the context of different types of pain in horses.** Horses with low grade orthopedic pain only show sporadic visual signs of pain, and these signs may overlap with spontaneous expressions of the non-pain class—it is therefore difficult to train a system solely on this data.

- We compare domain transfer from a horse pain dataset to standard transferred video features from a general large-scale action recognition dataset, and analyze whether these can complement each other.

- We present an explainability study of orthopedic pain detection in 25 video clips, firstly for a human expert baseline consisting of 27 equine veterinarians, secondly for one of our neural networks trained to recognize acute pain. We compare which signs of pain veterinarians typically look for when assessing horse pain with what the model finds important for pain classification.

    Next, we present related work in Section 2, followed by methodology along with dataset descriptions in Section 3. The emphasis lies on the experiments in Section 4 and the discussion thereof, presented in Section 5. Finally, we conclude and outline future directions in Section 6.

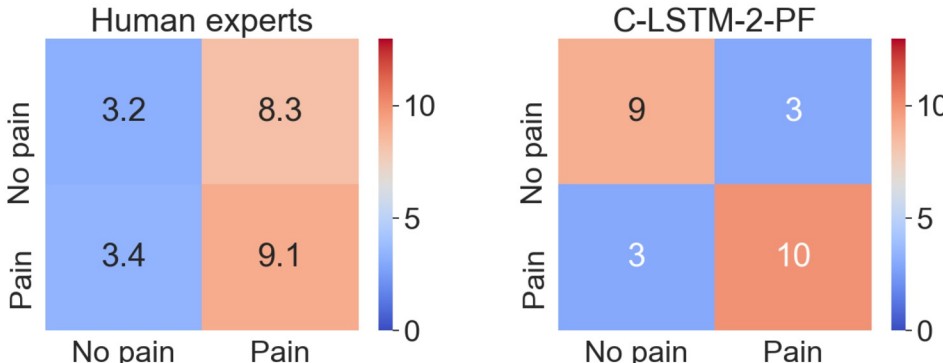

**Fig 2. Pain predictions on the 25 clips included in the baseline study (Table 2), by the human experts (*left*), and by the C-LSTM-2-PF † (*right*).**

## 2 Related work

Although many methods are relevant to our problem, our setting in terms of data is unique and requires a tailored approach. Weakly supervised action recognition is relevant in that we share the same goal: extracting pertinent information from weakly labeled video data. However, these methods typically rely on training with a large number of video clips, which is not accessible in our setting. Our orthopedic pain dataset consists of few (less than 100), although long (of several minutes) samples.

### 2.1 Weakly supervised action recognition and localization

Multiple-instance learning (MIL) has been used extensively within deep learning for the task of weakly supervised action localization (WSAL), where video level class labels alone are used to determine the temporal extent of class occurrences in videos [23–27]. However, training deep models within the MIL-scenario can be challenging. Error propagation from the instances may lead to vanishing gradients [28–30] and too quick convergence to local optima [31]. This is especially true for the low-sample setting, which is our case.

Typically, videos are split into shorter clips whose predictions are collated to obtain the video level predictions. Multiple methods use features extracted from a pre-trained two-stream model, I3D [32], as input to their weakly supervised model [23–25]. In addition to MIL, supervision on feature similarity or difference between clips in videos [23, 25, 33] and adversarial erasing of clip predictions [34, 35] are also used to encourage localization predictions that are temporally complete. In early stages of our study, we mainly attempted a MIL approach, but the noisy data, low number of samples and similar appearance of videos from the two classes were prohibitive for such a model to learn informative patterns.

Arnab et al. [36] cast the MIL-problem for video in a probabilistic setting, to tackle its inherent uncertainty. However, they rely on pre-trained detection of humans in the videos, which aids the action recognition. This is not applicable to our scenario since horses are always present in our frames (i.e., detecting a horse would not help us to temporally localize a specific behavior), and the behaviors we are searching for are more fine-grained than the human actions present in datasets such as [10] or [12] (e.g., fencing or eating). Another difference of WSAL compared to our setting is that we are agnostic as to what types of behavior we are looking for in the videos. For this reason, it is not possible for us to use a localization-by-temporal-alignment method such as the one by Yang et al. [37]. Moreover, their work relies on a small number of labeled and trimmed instances, which we do not have in this study.

### 2.2 Automatic pain recognition in animals

In [38], a Support Vector Machine (SVM) cascade framework is used to recognize facial action units in sheep from single images, to then assess pain according to pre-defined thresholds. A similar method is applied to horses and donkeys in [39], presenting per pain-related facial action unit classification results.

In [40], the work in [38] is continued, using automatically recognized sheep facial landmarks to assess pain on a single-frame basis. In this work, disease progression is monitored from video data, by applying the same pipeline on every 10th frame, and averaging their pain scores. Improving on [38], they use subject-exclusive (leave-one-animal-out) testing for the video part of their experiments.

Using a deep learning approach, Tuttle et al. [41] recognize induced inflammatory joint pain in albino mice from single images. While the classification task is binary, the pain labels are set according to human scorings based on facial expressions, on a scale from 1 to 10 (five action units associated with pain that each can have a confidence score of 0–2). It is not clear

from the article how they go from the ten-class scale to binary labels. Andresen et al. [42] apply this method to black-furred mice moving more freely in their cages compared to [41], still in a single-frame setting. Both methods [41, 42] use Imagenet [43] pre-trained, standard CNN networks [44, 45] as kept-fixed back-bones, while training a fully-connected classification head, on their datasets. Further, both can improve their classification accuracy by averaging the network confidence over images taken within a narrow time span. Andresen et al. [42] furthermore point to the difficulty of generalizing between different types of pain, which we investigate closer in this article.

Lencioni et al. [46] train light-weight CNN models (two convolutional layers) from scratch to recognize facial pain expressions automatically in horses from single images. Separate models are trained for the eye, ear and mouth regions, to recognize three levels of pain (0–2). The labels are entirely based on the Horse Grimace Scale (HGS), scored by humans, although the data was recorded before and after routine surgical castration. It is not clear if pain images were selected only post castration, or if the selection was made only based on visibility of the pain cues. Similarly, Li et al. [47] train separate CNN-based classifiers on small crops of different facial regions to recognize EquiFACS units [48], which can be used for pain evaluation.

In a previous work [22], we were the first to perform pain recognition in animals with models learning patterns from sequences rather than single frames, showing large improvement from training on single images. We used deep recurrent two-stream models, and trained with labels set according to clean experimental acute induced pain. The presented system uses no pre-defined behaviors or facial expressions, but learns spatio-temporal features based on raw videos and their pain labels only. In this article, we build on the same approach (with slight modifications listed in the Appendix of S1 File), and use it in this empirical study of how well different models can handle a domain shift in the test data.

In a more recent work of ours [49], we perform equine pain recognition on 3D pose representations extracted from multi-view surveillance data, on the same low grade orthopedic pain trial as in this paper. Although the horses and pain trial are the same as in the current work, the crucial difference is that the data used in [49] is different (surveillance data in the box, whereas here, we use videos recorded with a tripod outside the box, where the facial expression is visible), and that only the pose representation is used for classification. This is advantageous to reduce the amount of extraneous information. However, the potential disadvantage is that any facial expressions are not possible to take into account. As a result, it is perhaps the adjustment of pose as a result of previous pain that is recognized in [49], rather than whether a pain experience is ongoing. Similarly to the present work, it is found that low grade orthopedic pain is difficult to detect, compared to the less noisy pain trial used in [22].

### 2.3 Pain in horses

The definition of animal pain includes a change in motivation, where the animal develops behaviors to avoid pain or to protect the painful area [6]. Depending on the origin of pain, the animal may perform different behaviors. Horses with abdominal pain may stretch, roll and kick at the abdomen, while horses with orthopedic pain may be reluctant to move and have an abnormal weight distribution or movement pattern [7]. Therefore, pain assessment tools such as pain scales often target pain of a specific origin.

Facial expressions, on the other hand, seem to be universal for pain within a species. Grimace scales have been successfully applied to pain from different origins, such as post-surgical pain or laminitis in horses [17, 18, 50]. It also seems like pain-related facial expressions are present during both acute and chronic pain, and may be shown by the animal during several weeks post-injury, but not consistently and perhaps tailing away [51].

Acute pain has a sudden onset and a distinct cause, such as inflammation, while chronic pain is more complex, sometimes without a distinct cause, and by definition lasts for more than three months. Acute (and sometimes chronic) pain arises from the process of encoding an actually or potentially tissue-damaging event, so called nociception, and may therefore be referred to as nociceptive pain. When pain is associated to decreased blood supply and tissue hypoxia, it is instead termed ischemic pain [8, 9].

Orthopedic pain in horses can be of both acute and chronic character where a very common diagnosis is osteoarthritis, with related inflammatory pain of the affected joint [52]. In humans, the disease is known to initially result in nociceptive pain localized to the affected joint, but when chronic pain develops, central sensitization occurs with a more widespread pain [53]. How pain-related facial expressions and other behaviors vary between horses with acute and chronic orthopedic pain is yet to be described, and so is the relation between pain intensity and alterations in facial expressions.

## 3 Method

Central to this study are two datasets depicting pain of different origin in horses (Table 1). The datasets are similar in that they both show one horse, trained to stand still, either under pain induction or under baseline conditions (Section 3.1). In our experiments, we investigate the feasibility of knowledge transfer between different pain domains (Section 3.3). We use the macro average F1-score as metric, which is more conservative than accuracy when there is class imbalance—it does not favor the majority class.

### 3.1 Datasets

In Table 1, we show an overview of the two datasets used in this article. It can be noted that neither of the two show a full view of the legs of the horses. This could otherwise be an indicator of orthopedic pain. Both datasets mainly depict the face and upper body of the horses, see, e.g., Fig 3.

**3.1.1 The Pain Face dataset (PF).** The experimental setup and video recording of the PF dataset have been described in detail in [21, 22]. Briefly, the dataset consists of video recordings of six clinically healthy horses with and without acute pain. The pain is either ischemic (from a pressure cuff) or inflammatory (from a capsaicin substance on the skin) and was applied for short durations of time under ethically controlled conditions.

*3.1.1.1 Labels.* The induced pain is acute and takes place during 20 minutes, during which the horse shows signs of pain almost continuously. Thus, the video-level positive pain label is largely valid for all clips extracted from it. The labels are binary; any video clip extracted from this period is labelled as positive (1), and any video clip from a baseline recording is labelled as negative (0).

**3.1.2 The EquineOrthoPain(joint) dataset (EOP(j)).** The experimental setup for the EOP(j) dataset is described in detail in previously published work [16]. Mild to moderate orthopedic pain was induced in eight clinically healthy horses by injecting lipopolysaccharides

**Table 1. Overview of the datasets.** Frames are extracted at 2 fps, and clips consist of 10 frames. Duration shown in hh:mm:ss.

| Dataset | # horses | # videos | # clips | Pain | No pain | Total | Labeling |
|---|---|---|---|---|---|---|---|
| **PF** | 6 | 60 | 8784 | 03:41:25 | 06:03:44 | 09:45:09 | Video-level. Induced clean experimental acute pain, on/off (binary) |
| **EOP(j)** | 7 | 90 | 6710 | 03:37:36 | 05:03:25 | 08:41:01 | Video-level. Induced orthopedic pain, varying number of hours prior to the recording. Binarized human CPS pain scoring before/after recording (3 independent raters) |

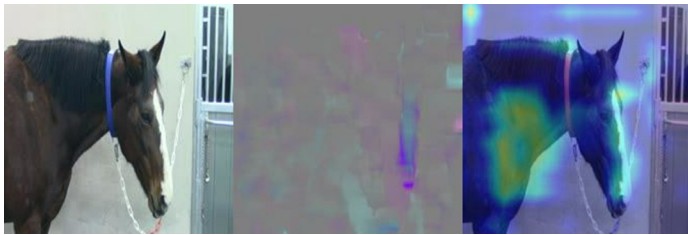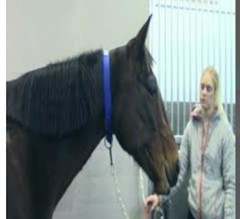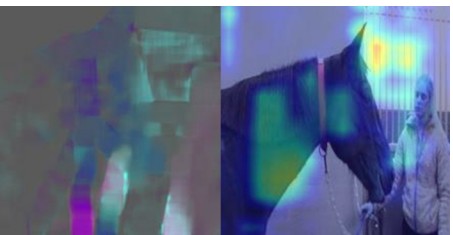

**Fig 3. The figures can be viewed as animations in the Supporting information.** Here, only the middle frame of each sequence is shown. RGB, optical flow, and Grad-CAM [62] saliency maps of the C-LSTM-2-PF † predictions on clips 10 and 24 (Table 2). Clip 10 (*left*) is a correct prediction of pain. Clip 24 (*right*) is a failure case, showing an incorrect pain prediction, and we observe that the model partly focuses on the human bystander. The remaining 23 clips with saliency maps can be found in the Appendix of S1 File.

(LPS) into the tarsocrural joint (hock). This is a well-known and ethically approved method for orthopedic pain induction in horses, resulting in a fully reversible acute inflammatory response in the joint [54]. Before, and during the 22–52 hour period after induction, several five minute videos of each horse were recorded regularly. A video camera, attached to a tripod at approximately 1.5 metres height and with 1.5 metres distance from the horse, recorded each horse when standing calmly in the stables outside the box stall.

*3.1.2.1 Labels.* The dataset contains 90 different videos associated with one pain label each (Table 1). Notably, these labels are set immediately before or after the recording of the video when the horse is in the box stall, and not simultaneously to the video recording. Three independent raters observed the horse, using the Composite Orthopedic Pain Scale (CPS) [55] to assign each horse a total pain score ranging from 0 to 39. The pain label is the average pain score of these three ratings. In this study, the lowest pain rating made was 0 and the highest was 10.

For the binary classification used in this study (following prior work [22]), the CPS score is thresholded so that any value larger than zero post-induction is labeled as pain, and values equal to zero are labeled as non-pain. This means that we consider possibly very weak pain signals (e.g., 0.33) as painful, adding to the challenging nature of the problem. One of the horses was excluded from our experiments, because it did not have CPS scores > 0 after the pain induction.

## 3.2 Cross-validation within one domain

When running cross-validation training, we train and test within the same domain. We train with leave-one-subject-out cross-validation. This means that one horse is used as validation set for model selection, one horse as held-out test set, and the rest of the horses are used for training. The intention with these experiments is to establish baselines and investigate the treatment of weak labels as dense labels for the two datasets.

**3.2.1 Treating weak labels as dense labels.** We distinguish between *clips* and *videos*, where clips are five second long windows extracted from the videos (several minutes long) (Table 1). For both datasets, the pain labels have been set weakly on video level. In practice, treating these labels as dense means giving the extracted clips the same label as the video.

**3.2.2 Architectures.** We use two models in our experiments: the two-stream I3D, pretrained on Kinetics (kept fixed until the 'Mixed5c' layer), and the recurrent convolutional two-stream model (hereon, C-LSTM-2) from [22]. Each stream (RGB and optical flow) of C-LSTM-2 consists of four blocks of convolutional LSTM-layers, with max pooling and batch normalization in each. The convolutional LSTM layer was first introduced by Shi et al. [56],

and replaces the matrix multiplication transforms of the classical LSTM equations (cf., [57]) with convolutions. This allows the layer to ingest data with a spatial grid structure, and to maintain a spatial structure for its output as well. The classical LSTM requires flattening all input data to 1D vectors, which is suboptimal for image data, where the grid structure matters. The two streams are fused by addition after the last layer, flattened and input to a two-class classification head. The classification head provided with the I3D implementation is kept and retrained to two classes.

The output of the models are binary pain predictions. We follow the supervised training protocol of [22] with minor modifications; more details can be found in the Appendix (Section C.1) of S1 File. The main difference is that we resample the minor class clips with a different window stride, to reduce the class imbalance. We also run on a higher frame resolution, 224x224 instead of 128x128. Further implementation details can be found in [22], in the Appendix of S1 File and in the public code repository.

### 3.3 Domain transfer

When running domain transfer experiments, we use two different methods. The first is to train a model on the entire dataset from one domain for a fixed number of epochs without validation, and test the trained model on another domain. This means that the model has never observed the test data domain. We also run experiments where we first pre-train a model on the source domain, and fine-tune its classification layer on the target domain. In this way, the model has acquainted itself with the target domain, but not seen the specific test subject.

To choose the number of epochs for model selection when training on the entire dataset, we use the average best number of epochs from when running intra-domain cross-validation (Section 3.2) and multiply this with a factor of how much larger the dataset becomes when including test and validation set (1.5 when going from 4 to 6 horse subjects). This was $77 * 1.5 = 115$ epochs when training C-LSTM on PF, and $42 * 1.5 = 63$ epochs when training I3D on PF. This takes around 80h on a GeForce RTX 2080 Ti GPU for the C-LSTM-2, which is trained from scratch, and around 4h for the I3D where only the classification head is trained. Except for the number of epochs, the model is trained with the same settings as during intra-domain cross-validation.

### 3.4 Veterinary expert baseline experiment

As a baseline for orthopedic pain recognition, we engaged 27 Swedish equine veterinarians in rating 25 clips from the EOP(j) dataset. In veterinary practice, the decision of whether pain is present or not is often made quickly based on the veterinarian's subjective experience The veterinarians were instructed to perform a rating of pain intensity of the horses in the clips using their preferred way of assessment. We asked for the intensity to be scored subjectively from no-pain (0) to a maximum of 10 (maximal pain intensity). The maximum allowed time to spend in total was 30 minutes. The average time spent was 18 minutes (43 seconds per clip). The participants were carefully instructed that there were clips of horses without and with pain, and that only 0 represented a pain-free state.

There is no gold standard for assessment of pain in horses. Veterinary methods rely on subjective evaluation of information collected on the history of the animal, its social interaction and attitude, owners' complaints and an evaluation of both physical examination and behavioral parameters [58]. In this case only the behavioral changes could be seen. The assessments in practice are rarely blinded, but influenced by knowledge of the history and physiological state of the animal or the observation of an obvious pain behaviours. To simulate the short time span for pain estimation and avoid expectation bias, each clip was blinded for all external

**Table 2. Overview of the predictions on 25 EOP(j) clips made by the human veterinarian experts and by one C-LSTM-2 instance, trained only on PF.** The labels for the C-LSTM-2 were thresholded above 0 (same threshold as for the experts). The behavior symbols in the Behavior column are explained in Table 3.

| Clip | Behaviors | CPS | Label | 27 Experts | | C-LSTM-2-PF † | |
| --- | --- | --- | --- | --- | --- | --- | --- |
| | | | | Avg. rating | # correct | Pred. | Conf. |
| 1 | e1 | 2 | 1 | 2.7 | 23 | 1 | 0.9999 |
| 2 | o1 l | 2 | 1 | 1.1 | 10 | 1 | 0.9241 |
| 3 | e2 t1 c1 n1 p | 4.33 | 1 | 3.4 | 22 | 1 | 0.9997 |
| 4 | e2 o1 l | 4.67 | 1 | 3.7 | 21 | 1 | 0.6036 |
| 5 | t1 c1 | 3.67 | 1 | 0.37 | 4 | 1 | 0.9853 |
| 6 | | 1.33 | 1 | 0.96 | 13 | 1 | 0.5200 |
| 7 | u | 4.33 | 1 | 1.3 | 12 | 1 | 0.9063 |
| 8 | c2 n2 m u p | 6.33 | 1 | 4.7 | 25 | 0 | 0.8623 |
| 9 | e1 t1 c1 n1 p | 3 | 1 | 4.6 | 25 | 0 | 0.5504 |
| 10 | e2 t1 c1 n2 l p | 2 | 1 | 4.4 | 25 | 1 | 0.9999 |
| 11 | e2 t1 c1 n2 l | 1 | 1 | 6.8 | 27 | 1 | 0.8231 |
| 12 | e1 t1 c1 n1 m | 4.33 | 1 | 2.5 | 22 | 0 | 0.8046 |
| 13 | e2 t1 c1 n2 | 1.67 | 1 | 4.4 | 26 | 1 | 0.9840 |
| 14 | e2 o1 t1 c1 n1 u | 0 | 0 | 5.5 | 0 | 0 | 0.9993 |
| 15 | e2 t1 | 0 | 0 | 4.1 | 1 | 0 | 0.5848 |
| 16 | e2 o1 t1 c1 n1 | 0 | 0 | 3.9 | 4 | 0 | 0.8439 |
| 17 | e1 o1 t1 n1 l p | 0 | 0 | 4.4 | 5 | 1 | 0.6456 |
| 18 | t1 m u p | 0 | 0 | 0.96 | 17 | 0 | 0.9991 |
| 19 | | 0 | 0 | 0.48 | 21 | 0 | 0.9568 |
| 20 | t1 l | 0 | 0 | 2.9 | 9 | 1 | 1.00 |
| 21 | u | 0 | 0 | 1.3 | 13 | 0 | 0.9999 |
| 22 | | 0 | 0 | 2.7 | 4 | 0 | 0.6128 |
| 23 | c1 n1 u | 0 | 0 | 1.6 | 8 | 0 | 0.9989 |
| 24 | e1 t1 n1 h p | 0 | 0 | 2.4 | 9 | 1 | 1.00 |
| 25 | e2 o1 t1 n1 | 0 | 0 | 5.9 | 0 | 0 | 0.6135 |

information, and only five seconds long, i.e., the same temporal footprint as the inputs to the computer models. The intention with this was to keep the comparison to a computer system more pragmatic—a diagnostic system is helpful to the extent that it is on par with or more accurate than human performance, reliable *and* saves time. Another motivation to use short clips was for the feasibility of the study, and to avoid rater fatigue. It is extremely demanding for a human to maintain subtle behavioral cues in the working memory for longer than a few seconds at a time. In effect, this fact in itself pinpoints the need for an automatic pain recognition method.

The clips selected for this study were sampled from a random point in time from 25 of the videos of the EOP(j) dataset, 13 pain and 12 non-pain. Only pain videos with a CPS pain label $\geq 1$ were included in order to have a clearer margin between the two classes, making the task slightly easier than on the entire dataset. We first extracted five such clips from random starting points in the video, and used the first of those where the horse was standing reasonably still without any obstruction of the view.

Behaviors in the 25 clips were manually identified and listed in Table 2. Those related to the face were identified by two other veterinary experts in consensus according to the Horse Grimace Scale [17]. The behaviors we were attentive to for each clip are listed in Table 3.

**Table 3. Explanation of the listed behavior symbols appearing in Table 2.**

| Behavior | Symbol |
|---|---|
| *From the* Horse Grimace Scale [17] | |
| Backwards ears, moderately present | e1 |
| Backwards ears, obviously present | e2 |
| Orbital tightening, moderately present | o1 |
| Orbital tightening, obviously present | o2 |
| Tension above the eye area, moderately present | t1 |
| Tension above the eye area, obviously present | t2 |
| Mouth strained and pronounced chin, moderately present | c1 |
| Mouth strained and pronounced chin, obviously present | c2 |
| Strained nostrils and flattening of the profile, moderately present | n1 |
| Strained nostrils and flattening of the profile, obviously present | n2 |
| *Other* | |
| Large movement | m |
| Mouth play | p |
| Lowered head | l |
| Clearly upright head | u |
| Human in clip | h |

# 4 Experiments

In this section, we describe our results from intra-domain cross-validation training (4.1), domain transfer (4.2) and from the human expert baseline study on EOP(j) and its comparison to the best performing model, which was trained only on acute pain (4.3).

## 4.1 Cross-validation within one domain

The results from training with cross-validation within the same dataset are presented in Table 4 for both datasets. When training solely on EOP(j), C-LSTM-2 could not achieve a higher result than random performance (49.5% F1-score), and I3D was just above random (52.2). Aiming to improve the performance, we combined the two datasets in a large 13-fold training rotation scheme. After mixing the datasets, and thereby almost doubling the training set size and number of horses, the total results on 13-fold cross-validation for the two models were 60.2 and 59.5 on average, but where the PF folds on average obtained 69.1 and 71.3 and the EOP(j) folds obtained 53.4 and 49.4. Thus, the performance on PF deteriorated for both models, as well as on EOP(j) for I3D (49.4) and only slightly improved on EOP(j) (53.4) for C-LSTM-2. This indicates that the weak labels of EOP(j) and general domain differences between the datasets hindered standard supervised training with a larger, combined dataset.

## 4.2 Domain transfer to EOP(j)

Table 5 compared to Table 4 shows the importance of domain transfer for the task of recognizing pain in EOP(j). One trained instance of the C-LSTM-2, which has never seen the EOP(j) dataset (hereon, C-LSTM-2-PF †), achieves 58.2% F1-score on it—higher than any of the other approaches. I3D, which achieved higher overall score when running cross-validation on PF, did not generalize as well to the unseen EOP(j) dataset (52.7). For I3D, trials with models trained during a varying number of epochs are included in the Appendix, although none performed better than 52.7% F1-score.

**Table 4. Results (% F1-score) for intra-domain cross-validation for the respective datasets and models.** The results are averages of five repetitions of a full cross-validation and the average of the per-subject-across-runs standard deviations.

| Dataset | # horse folds | F1-score | Accuracy |
|---|---|---|---|
| **C-LSTM-2** | | | |
| PF | 6 | 73.5 ±7.1 | 75.2 ±7.4 |
| EOP(j) | 7 | 49.5 ±3.6 | 51.2 ±2.8 |
| PF + EOP(j) | 13* | 60.2 ±2.6 | 61.8 ±3.2 |
| (*PF | | 69.1 ±4.9 | 71.1 ±3.9) |
| (*EOP(j) | | 53.4 ±3.0 | 53.9 ±2.5) |
| **I3D** | | | |
| PF | 6 | 76.1 ±1.5 | 76.6 ±1.1 |
| EOP(j) | 7 | 52.2 ±2.3 | 52.6 ±2.2 |
| PF + EOP(j) | 13* | 59.5 ±4.3 | 62.2 ±2.7 |
| (*PF | | 71.3 ±3.5 | 73.1 ±1.4) |
| (*EOP(j) | | 49.4 ±5.3 | 52.9 ±3.7) |

Fine-tuning (designated by FT in Table 5) these PF-trained instances on EOP(j) decreased the result (54.0 and 51.8, respectively), presumably due to the lesser amount of clearly discriminative visual cues in the EOP(j) data. This goes in line with results in Table 4; the data and labels of the EOP(j) data do not seem to be suited for supervised training.

Table 6 shows that the results for individual horses may increase when applying a multiple-instance learning filter during inference to the predictions across a video and base the classification only on the top 1%/5% confident predictions (significantly for subjects A, H, and I, and slightly for J and K); however, for other subjects, the results decreased (B, N). As described in Section 4.3, there may be large variations among individuals for this type of pain induction.

## 4.3 Veterinary expert baseline experiment

The method of the expert baseline study is described in Section 3.4. Next, we compare and interpret the decisions of the human experts and the C-LSTM-2-PF † on the 25 clips of the study.

**4.3.1 Comparison between the human experts and the C-LSTM-2-PF †.** Table 2 gives an overview of the ratings of the 25 clips given by the experts and by the model. First, we note that the C-LSTM-2-PF † instance outperforms the humans on these clips, achieving 76.0%

**Table 5. F1-scores on EOP(j), when varying the source of domain transfer, for models trained according to Section 3.3.** FT means fine-tuned (three repetitions of full cross-validation runs). Column letters indicate different test subjects. † represents a specific model instance, reoccurring in Tables 2, 6 and 7.

| Model | A | B | H | I | J | K | N | Global |
|---|---|---|---|---|---|---|---|---|
| **I3D** [32], Kinetics | | | | | | | | |
| PF, EOP(j)-FT | 48.7±4.1 | 56.8±0.9 | 47.0±3.4 | 43.6±4.7 | 53.6±1.7 | 54.5±0.9 | 58.2±0.6 | 51.8±2.3 |
| PF, EOP(j) unseen | 54.96 | 45.05 | 53.42 | 56.52 | 55.69 | 41.66 | 46.59 | 52.70 |
| 3 repeated runs | 49.9 ±2.8 | 48.0 ±2.1 | 47.2 ±2.6 | 52.2 ±1.8 | 59.0 ±1.0 | 40.7 ±0.8 | 46.1 ±2.2 | 52.6 ±0.4 |
| **C-LSTM-2**, scratch | | | | | | | | |
| PF, EOP(j)-FT | 50.4±2.7 | 55.1±0.8 | 44.9±1.4 | 45.7±0.7 | 69.3±1.7 | 51.2 ±0.7 | 60.8±0.8 | 54.0±1.3 |
| PF, EOP(j) unseen † | 61.55 | 56.34 | 55.55 | 51.51 | 64.76 | 45.11 | 57.84 | 58.17 |
| 3 repeated runs | 59.4 ±4.6 | 56.7 ±2.7 | 60.5 ±5.7 | 52.1 ±1.8 | 53.2 ±14.0 | 49.6 ±3.9 | 54.2 ±4.2 | 56.3 ±2.8 |

**Table 6. Results on video-level for EOP(j), when applying a multiple-instance learning (MIL) filter during inference on the clip-level predictions.** The column letters designate different test subjects. The model has never trained on EOP(j), and is the same model instance as in Tables 2, 5 and 7.

| Model instance | MIL-filter | A | B | H | I | J | K | N |
|---|---|---|---|---|---|---|---|---|
| C-LSTM-2-PF † | - | 61.55 | 56.34 | 55.55 | 51.51 | 64.76 | 45.11 | 57.84 |
| C-LSTM-2-PF † | Top 5% | 88.31 | 51.13 | 59.06 | 59.06 | 65.37 | 31.58 | 36.36 |
| C-LSTM-2-PF † | Top 1% | 88.31 | 51.13 | 53.33 | 59.06 | 65.37 | 45.83 | 45.0 |

**Table 7. F1-scores (%) on the 25 clips of the expert baseline.** The C-LSTM-2-PF † instance was trained on PF but never on EOP(j). Asterisk: results on the entire EOP(j) dataset for comparison.

| Rater | No pain | Pain | Total |
|---|---|---|---|
| Human expert | 34.7 ±10.0 | 60.6 ±4.4 | 47.6 ±5.5 |
| C-LSTM-2-PF † | 75.0 | 76.9 | 76.0 |
| C-LSTM-2-PF †* | 54.47 | 61.86 | 58.17 |

F1-score, compared to 47.6 ± 5.5 for the experts (Table 7, Fig 2). The experts mainly had difficulties identifying non-pain-sequences. Similarly, however, when the model was tested on the entire EOP(j) dataset, its non-pain results were lower than its pain results as well (Table 7), indicating the difficulty of recognizing the non-pain category.

Most of the clips rated as painful by the experts contain behaviors that are classically associated with pain, for example as described in the Horse Grimace Scale [59]. Among the pain clips, clip 6 is the only one without any listed typically pain-related behaviors. The veterinarians score the clip very low (0.96) and seem to agree that the horse does not look painful. The model interestingly scores the clip as painful, but with a very low confidence (0.52).

There are three clips with clear movement of the horse (8, 12, 18), where 8 and 12 are wrongly predicted by the model as being non-pain. Clip 18 is correctly predicted as being non-pain with high confidence (0.9991), suggesting that the model associates movement with non-pain. On clip 18, the human raters mostly agree (17) with the model that this horse does not look painful and the average rating is low (0.96). It can further be noted that the three incorrect pain predictions (17, 20, 24) made by the model occurred when there was either e1 (moderately backwards ears, pointing to the sides) or l (lowered head), or both. Also, 24 is the only clip with a human present, which might have confused the model further (Fig 3).

The four most confident and correct non-pain predictions (>0.99) made by the model are the ones where the head is held in a clear, upright (u) position. Similarly, the three most confident and correct pain predictions (>0.99) by the model all contain ear behavior (e1 or e2).

## 5 Discussion

### 5.1 Why is the expert performance so low?

Tables 7 and 8 and Fig 2 show a low performance for the human experts in general and especially for non-pain.

Increasing the threshold to 1 and 2 reduced the accuracy for pain, which may be due to false inclusion of scores of 0 if the raters scored 1 or 2 for non-pain, contrary to the instructions. Vice versa, the accuracy for non-pain increased when the threshold was extended, which may be due to inclusion of scores of 1 and 2, used as non-pain (even though they were informed that only 0 is used for non-pain). A reluctance to assess zero pain is difficult for clinicians who are taught that signs of pain may be subtle.

**Table 8. Accuracies (%) from the expert baseline, varying with the chosen pain threshold.**

| Threshold | No pain | Pain | Total |
|-----------|---------|------|-------|
| 0 | 28.1 ±11.2 | 72.7 ±9.4 | 51.3 ±4.6 |
| 1 | 37.4 ±14.1 | 65.2 ±9.7 | 51.9 ±5.7 |
| 2 | 48.5 ±18.7 | 52.7 ±13.8 | 50.7 ±7.5 |

The results point to the difficulty of observing pain expressions at a random point in time for orthopedic pain, and without context. The LPS-induced orthopedic pain may further have complicated the rating process, since it varies in intensity among individuals, despite administration of the same dose. This results in different levels of pain expressions [60], sometimes occurring intermittently. Hence, there will be 'windows' during the observed time where the horse expresses pain clearly [61]. The other parts of the observed time will then contain combinations of facial expressions that some raters interpret as non-pain, and some raters interpret as pain. If a 'window' is not included in the five second clip, it is difficult for the rater to assign a score, decreasing their accuracy.

## 5.2 Significance of results

Having trained the C-LSTM-2 on a cleaner source domain (PF), without ever seeing the target domain (EOP(j)) before, gave better results than all other attempts, including fine-tuning (58.2% F1-score for the best instance, and 56.3±2.8% for three repeated runs). Despite being higher than human performance, these F1-scores on the overall dataset are still modest and significantly lower than the recognition of acute pain in [22]. However, the results are promising, especially since they were better for the clips used for the human study (with higher pain-scores) (76%, vs. 48% for the human experts). This may mean that the noise in the labels on the overall dataset—both inherent to pain labelling and specific for the sparse pain behavior related to low grade orthopedic pain, obscures the system's true performance to some extent.

The human expert baseline for classification on clip-level of the EOP(j) dataset, together with the intra-domain results (Table 4), shows the difficulty in detecting orthopedic pain for humans and standard machine learning systems trained in a supervised manner, within one domain. Poor performance of raters in assessing low grade pain is the case generally, and points to the necessity of this study. The lack of consensus is troubling since veterinary decision-making regarding pain recognition is critical for the care of animals in terms of prescribing alleviating treatments and in animal welfare assessments [63]. As an example of this, veterinarians can score assumed pain in horses associated with a particular condition on a range from 'non-painful' to 'very painful' [64]. One important advantage of an automatic pain recognition system would be its ability to store information over time, and produce reliable predictions according to what has been learned previously. Humans are not able to remember more than a few cues at the time when performing pain evaluation. This creates the need for automated methods and prolonged observation periods, where automated recognition can indicate possible pain episodes for further scrutiny. In equine veterinarian clinics, such a system would be of great value. In summary, even a system with a less-than-perfect accuracy would be useful in conjunction with experts on site.

## 5.3 Expected generalization of results

This study has been performed on a cohort of, in total, $n = 13$ horses. It is therefore, as always, important to bear in mind the possible bias in these results. Nevertheless, we want to

emphasize that the paper was dedicated to investigating generalizability, and that there already is a domain gap between the two groups of $n = 6$ and $n = 7$ horses. The recordings of the two groups (datasets) were made four years apart, in different countries, and, naturally, on entirely different horse subjects. In addition to this, whenever we evaluated our system in the intra-domain setting, the test set consisted only of data from a previously unseen individual (leave-one-subject-out testing). Considering this, our findings do indicate that the method would generalize to new individuals—in particular if the system could be trained on an increased amount of clean base-domain data.

## 5.4 Differences in pain biology and display of pain in PF and EOP(j)

Both video sets were recorded of horses under short term acute pain after a base line period. However, the noxious stimuli and the anatomical location of the pain differed widely. The PF dataset was created by application of two well-known experimental noxious stimuli of only little clinical relevance (capsaicin [65] and ischemia [66]). Both stimuli are used in in pain research in human volunteers, induce pain lasting for 10–30 minute and pain levels corresponding to 4 or 5 on a 10 point scale, where 0 is no pain and 10 is worst imaginable pain. Due to this short time span, the controlled course of pain intensity, the controlled experimental conditions and the predictability of the model, these data present the most noise-less display of possible behavioural changes due to the pain experienced. Further, because the pain is of such short duration, the horse will not be able to compensate or modify its behaviours. However, such data are less useful for clinical situations. During clinical conditions, pain intensity is unpredictable, intermittent and of longer duration, allowing the horse to adapt to the pain, according to its previous experience and temperament. In real clinical situations, there is no ground truth of the presence or intensity of pain. The LPS model represents an acute, joint pain caused by inflammation of the synovia, resulting in orthopedic pain which ceases within 24 hours. The degree and onset of inflammation, and thus the resulting pain is known to be individual, depending on a range of factors which can not be accounted for in horses, including immunological status and earlier experiences with pain [67]. Because the horse has time to to adapt to and compensate the pain, by for example unloading the painful limb, pain will be intermittent or of low grade presenting in unpredictable epochs [68]. The low-noise data set therefore showed to be feasible to learn from, even if the pain kinds were different. Whether a low noise dataset also can improve recognition of chronic or neuropathic pain types, remains to be investigated.

## 5.5 Pain intensity and binarization of labels

As noted above, the labels in the PF dataset were set as binary from the beginning, according to whether the pain induction was ongoing or not, while the binary labels in the EOP(j) dataset were assigned afterwards, based on thresholding of the raters' CPS scores. The videos in EOP(j) were recorded during pain progression and regression. Hence, they contain different pain intensities, ranging from very mild to moderate pain. Introducing more classes in the labeling may mirror the varying intensities more accurately than binary labels, but the low number of samples in EOP(j) (90) restricts us to binary classification. Increasing the number of classes would not be sound in this low-sample scenario, when using supervised deep learning for classification, a methodology which relies on having many samples per class, in order to learn patterns statistically.

Furthermore, deciding pain intensity labels in animals is difficult. More accurate human pain recognition has been found for higher grimace pain scores [69], underlining that mild pain intensity is challenging to assess. This is in agreement with studies in human patients,

where raters assessing pain-related facial expressions struggled when the patients reported a mild pain experience [70]. Grimace scores seem to follow the regression of pain after analgesic administration [71] and may therefore aid in defining pain intensity. However, the relation between pain intensity and level of expression is known to be complex in humans and may be so in animals. Pain intensity estimation on a Visual Analogue Scale was not accurate enough in humans, and the estimation seemed to benefit from adding pain scores assessing pain catastrophizing, life quality and physical functioning [72]. As discussed by [63], pain scores may instead be used to define the likelihood of pain, where a high pain score increases the likelihood that the animal experiences pain. In addition, when pain-related behaviors were studied in horses after castration, no behaviors were associated to pain intensity [73]. This leaves us with no generally accepted way to estimate pain intensity in animals, supporting our choice of using binary labels in this study.

## 5.6 Labels in the real world

None of the equine pain datasets were recorded with the intention to run machine learning on the videos. This presents noise, in both data and labels. We point to how one can navigate a fine-grained classification problem, on a real-world dataset in the low-data regime, and show empirically that knowledge could be transferred from a different domain (for the C-LSTM-2 model), and that this was more viable than training on the weak labels themselves.

## 5.7 Domain transfer: Why does the C-LSTM-2 generalize better than I3D?

Despite performing better on PF during intra-domain cross-validation, I3D does worse upon domain transfer to a new dataset (Table 5) compared to the C-LSTM-2. It is furthermore visible in Table 4 that the I3D performance on EOP(j) deteriorates when combining the two datasets, perhaps indicating a proneness to learning dataset-specific spurious correlations which do not generalize. In contrast, the C-LSTM-2 slightly improves its performance on EOP(j) when merging the two training sets.

We hypothesize that this is because I3D is an over-parameterized model (25M parameters), compared to the C-LSTM-2 (1.5M parameters). An I3D pre-trained on Kinetics with its large number of trainable parameters is excellent when a model needs to memorize many, predominantly spatial, features of a large-scale dataset with cleanly separated classes, in an efficient way. When it comes to fine-grained classification of a lower number of classes, which can generalize to a slightly different domain, and moreover requires more temporal modeling than when the task is to separate 'playing trumpet' from 'playing violin' (or at Kinetics' most challenging: 'dribbling' from 'dunking'), it seems, from our experiments, that it is not a suitable architecture.

Another reason could be the fact that the C-LSTM-2 is trained solely on horse data, from the bottom up, while the I3D has its back-bone unchanged in our experiments. In that light, the C-LSTM-2 can be considered more specialized to the problem. Although Kinetics-400 does contain two classes related to horses: 'grooming horse' and 'riding or walking with horse', the C-LSTM-2 undoubtedly has seen more up-close footage of horses. In fact, somewhat ironically, the 'riding or walking with horse' coupled with 'riding mule' is listed in [11] as the top confused class of the dataset, using the two-stream I3D.

How does I3D do if trained solely on the PF dataset? This is where the model size becomes a problem. I3D requires large amounts of training data to converge properly; the duration of Kinetics-400 is around 450h. It is, for ethical reasons, difficult to collect a 450h video dataset (>40 times larger than PF) with controlled pain labels. Table 9 shows additional results when training I3D either completely from scratch (random initialisation) on the PF data, or from a

**Table 9. Global average F1-scores for domain transfer experiments for I3D, using varying pre-training and fine-tuning schemes.** The model is trained on the PF dataset and tested on the EOP(j) dataset. Only the pre-trained model, fine-tuned with a frozen back-bone could achieve results slightly above random performance on EOP(j).

| Epoch | Scratch | Pre-trained | Pre-trained freeze back-bone |
|---|---|---|---|
| 25 | 46.8 ±8.9 | 46.5 ±4.4 | 51.7 ±1.4 |
| 63 | 46.3 ±9.3 | 46.4 ±7.5 | 52.6 ±0.4 |
| 115 | 46.6 ±10.4 | 47.4 ±1.4 | 52.6 ±0.05 |
| 200 | 46.3 ±7.2 | 43.2 ±3.8 | 52.4 ±0.5 |
| # trainable parameters | 24,542,116 | 24,542,116 | 4,100 |

pre-trained initialisation, compared to when only training the classification head (freezing the back-bone). The results point to the difficulty of training such a large network in the low data regime.

## 5.8 Weakly supervised training on EOP(j)

During the course of this study, we performed a large number of experiments in a weakly supervised training regime on EOP(j). Our approach was to extract features from pre-trained networks and combine these into full video-length, to then run multiple-instance learning training on the feature sequences (the assumption being that a pain video would contain many negative instances as well). The training was attempted using both simple fully-connected networks, LSTM models and attention-based Transformer models. Training on the full video-length is computationally feasible since the features are low-dimensional compared to the raw video input. The predictions from the pre-trained networks were also used in this training scheme, both as attention within a video-level model or as pseudo-labels when computing the various loss functions we experimented with.

The results were generally not higher than random, even when re-using the features from the best performing model instance (C-LSTM-2-PF †). Our main obstacle, we hypothesize, was the low statistical sample size (90) on video-level. To run weakly supervised action or behavior recognition, a large number of samples, simply a lot of data, is needed—otherwise the training is not stable. This was visible from the significant variance across repeated runs in this type of setting. Controlled video data of the same horse subjects, in pain and not, does not (and, for ethical reasons, should not) exist in abundance. For this reason, we resorted to domain transfer from clean, experimental acute pain as the better option for our conditions.

## 6 Conclusions

We have shown that domain transfer is possible between different pain types in horses. This was achieved through experiments on two real-world datasets presenting significant challenges from noisy labels, low number of samples and subtle distinction in behavior between the two classes.

We furthermore described the challenges arising when attempting to move out of the cleaner bench-marking dataset realm, which is still under-explored in action recognition. Our study indicated that a deep state-of-the-art 3D convolutional model, pre-trained on Kinetics, was less suited for fine-grained action classification of this kind than a smaller convolutional recurrent model which could be trained from scratch on a clean source domain of equine acute pain data.

A comparison between 27 equine veterinarians and a neural network trained on acute pain was conducted on which behaviors were preferred by the two, respectively. The comparison indicated that the neural network prioritized other behaviors than humans during pain-non-

pain classification. We thus demonstrated that the domain transfer may function better for low grade pain recognition than human expert raters, when the classification is pain-no pain.

We presented the first attempt at recognizing low grade orthopedic pain from raw video data and hope that our work can serve as a stepping stone toward further recognition and analysis of horse pain behavior in video.

### 6.1 Future work

Directions for future work include processing data showing the horse in a more natural environment, such as in its box or outdoors, among other horses, though it might be challenging to collect data in these circumstances. This would require a more robust tracking of the horse in the video, for instance using animal pose estimation methods such as [74, 75]. Learning to discriminate between other affective states, such as stress and pain, or the opposite, recognizing when an animal is free of pain, is another important but difficult avenue to consider [63, 76].

## Supporting information

**S1 File.**
(ZIP)

**S2 File.**
(ZIP)

## Acknowledgments

The authors would like to thank Elin Hernlund and Marie Rhodin for valuable discussions.

## Author Contributions

**Conceptualization:** Sofia Broomé, Katrina Ask, Maheen Rashid-Engström, Pia Haubro Andersen, Hedvig Kjellström.

**Data curation:** Sofia Broomé, Katrina Ask.

**Funding acquisition:** Pia Haubro Andersen, Hedvig Kjellström.

**Investigation:** Sofia Broomé, Katrina Ask.

**Methodology:** Sofia Broomé, Katrina Ask, Maheen Rashid-Engström.

**Project administration:** Sofia Broomé.

**Software:** Sofia Broomé.

**Supervision:** Pia Haubro Andersen, Hedvig Kjellström.

**Visualization:** Sofia Broomé.

**Writing – original draft:** Sofia Broomé, Katrina Ask, Pia Haubro Andersen, Hedvig Kjellström.

**Writing – review & editing:** Sofia Broomé, Katrina Ask, Maheen Rashid-Engström, Pia Haubro Andersen, Hedvig Kjellström.

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
