## [Decision Letter · Decision Letter 0]

28 Jan 2022

Sharing Pain: Using Pain Domain Transfer for Video Recognition of Low Grade Orthopedic Pain in Horses

PONE-D-21-40066

Dear Dr. Broome,

We’re pleased to inform you that your manuscript has been judged scientifically suitable for publication and will be formally accepted for publication once it meets all outstanding technical requirements.

Kind regards,

Ayan Seal, Ph.D

Academic Editor

PLOS ONE

Reviewers' comments:

Reviewer #1: The manuscript was intelligently written and the statistical analysis was rigorously done but the data isn't enough to come to conclusion for the research. It is understandable that is challenging to collect enough data to conduct the research. But a great job was done.

Reviewer #2: Presented article consists an interesting work. Authors have worked hard for statistical analysis and has been performed appropriately and rigorously. Article is recommended for accept in current form.

Reviewer's Responses to Questions

**Comments to the Author**

1. Is the manuscript technically sound, and do the data support the conclusions?

Reviewer #1: No

Reviewer #2: Yes

2. Has the statistical analysis been performed appropriately and rigorously? 

Reviewer #1: Yes

Reviewer #2: Yes

3. Have the authors made all data underlying the findings in their manuscript fully available?

Reviewer #1: No

Reviewer #2: Yes

4. Is the manuscript presented in an intelligible fashion and written in standard English?

Reviewer #1: Yes

Reviewer #2: Yes

5. Review Comments to the Author

Reviewer #1: The manuscript was intelligently written and the statistical analysis was rigorously done but the data isn't enough to come to conclusion for the research. It is understandable that is challenging to collect enough data to conduct the research. But a great job was done.

Reviewer #2: Presented article consists an interesting work. Authors have worked hard for statistical analysis and has been performed appropriately and rigorously. Article is recommended for accept in current form.

6. PLOS authors have the option to publish the peer review history of their article (what does this mean?). If published, this will include your full peer review and any attached files.

Reviewer #1: No

Reviewer #2: **Yes: **Krishna Kumar Sharma

---

## [Editor Report · Acceptance letter]

23 Feb 2022

PONE-D-21-40066 

Sharing Pain: Using Pain Domain Transfer for Video Recognition of Low Grade Orthopedic Pain in Horses 

Dear Dr. Broomé:

I'm pleased to inform you that your manuscript has been deemed suitable for publication in PLOS ONE. Congratulations! Your manuscript is now with our production department. 

Kind regards, 

on behalf of

Dr. Ayan Seal 

Academic Editor

PLOS ONE